# Genetic Identification and Drug-Resistance Characterization of *Mycobacterium tuberculosis* Using a Portable Sequencing Device. A Pilot Study

**DOI:** 10.3390/antibiotics9090548

**Published:** 2020-08-27

**Authors:** Jorge Cervantes, Noemí Yokobori, Bo-Young Hong

**Affiliations:** 1Paul L. Foster School of Medicine, Texas Tech University Health Sciences Center, El Paso, TX 79905, USA; 2Servicio de Micobacterias, Instituto Nacional de Enfermedades Infecciosas (INEI)-ANLIS and CONICET, Buenos Aires C1282AFF, Argentina; kaoru.noemi@gmail.com; 3The Jackson Laboratory for Genomic Medicine, Farmington, CT 06032, USA; auvers@gmail.com

**Keywords:** next-generation sequencing, MinION, tuberculosis, drug resistance

## Abstract

Clinical management of tuberculosis (TB) in endemic areas is often challenged by a lack of resources including laboratories for *Mycobacterium tuberculosis* (Mtb) culture. Traditional phenotypic drug susceptibility testing for Mtb is costly and time consuming, while PCR-based methods are limited to selected target loci. We herein utilized a portable, USB-powered, long-read sequencing instrument (MinION), to investigate Mtb genomic DNA from clinical isolates to determine the presence of anti-TB drug-resistance conferring mutations. Data analysis platform EPI2ME and antibiotic-resistance analysis using the real time ARMA workflow, identified Mtb species as well as extensive resistance gene profiles. The approach was highly sensitive, being able to detect almost all described drug resistance conferring mutations based on previous whole genome sequencing analysis. Our findings are supportive of the practical use of this system as a suitable method for the detection of antimicrobial resistance genes, and effective in providing Mtb genomic information. Future improvements in the error rate through statistical analysis, drug resistance prediction algorithms and reference databases would make this a platform suited for the clinical setting. The small size, relatively inexpensive cost of the device, as well as its rapid and simple library preparation protocol and analysis, make it an attractive option for settings with limited laboratory infrastructure.

## 1. Introduction

Tuberculosis (TB) is the number one cause of human death due to an infectious disease, with 1.7 million deaths per year worldwide [1]. The causative agents of TB are a group of closely related bacteria known as the *Mycobacterium tuberculosis* (Mtb) complex (MTBC), which has been thought to have low DNA sequence diversity [2]. This limited diversity, however, is influenced by selective pressures and background selection [2]. Various human-adapted MTBC variants are known to differ in virulence, progression of disease and transmission potential.

TB surveillance of highly-virulent and multi-drug resistant (MDR) strains is paramount for adequate diagnosis and treatment [1,3]. Traditional phenotypic drug susceptibility testing (DST) through culture-based methods has multiple caveats, amongst them being that TB culturing can take days to weeks [4]. To reduce the time to obtain test results, alternative methods like real-time PCR-based Xpert MTB/RIF testing have been recommended by the World Health Organization [5]. These methods, however, are unable to detect drug-resistance mutations outside of the selected target loci [6], or they can produce false positive results [7]. In addition, clinical management where TB risk is high is often challenged by a lack of resources such as facilities for chest X-rays or laboratories for Mtb isolation and culture. To address these challenges, a whole-genome sequencing (WGS) approach can generate antibiotic susceptibility profiles, detect MDR-TB, and discover other MTB virulence factors [3,4]. This method, however, is also limited by resources, hospital–laboratory infrastructure and personnel training in bioinformatic analysis. A hybridization-based system (reverse line probe assay) has been recently proposed as an alternative in cost to WGS, but since this methodology is based on hybridization, it is also limited to the genomic region of Mtb examined [8]. Furthermore, although the cost per sample is much less than for other assays, it still requires laboratory equipment.

Development of a diagnostic assay that can be used at the point of care to rapidly and accurately diagnose TB and to include multidrug-resistant tuberculosis (MDR-TB) or extensively drug-resistant tuberculosis (XDR-TB) should be given a high priority. MDR-TB characterization typically requires costly machinery and handling in a specialized reference laboratory, not to mention the time required for shipping and processing the sample. A portable sequencing system that could be taken to the field, would not only reduce the cost of TB testing, but will also speed up the diagnoses. A rapid direct sample sequencing device would significantly reduce the time to obtain test results.

The MinION -Oxford Nanopore Technologies Limited (ONT), is a pocket sized (10 × 3 × 2 cm), portable, USB-powered, long-read sequencing instrument [9]. Among the existing sequencing platforms, it has the potential to be the best suited method to investigate the chain of transmission of TB and to determine the susceptibility of anti-TB drugs in the near future. This platform is particularly useful in remote settings or with limited infrastructure [9]. A careful evaluation of MinION as a potential methodology for the surveillance of TB was first reported in 2017 [10]. In this investigation, authors used both Illumina and ONT platforms for the diagnosis of Mtb infection. Utilization of the MinION in this study was conducted only with simulated Mtb infection using Ziehl–Neelsen (ZN)-negative sputum DNA combined with *Mycobacterium bovis* BCG strain DNA, not direct sputum sample. Despite the advantage of a portable sequencer in MDR-TB testing, so far there is no peer-reviewed published protocol of ONT-WGS based, rapid MDR-TB testing of patient sputum samples. It is unknown if this portable DNA sequencing system would be effective in providing information on Mtb genotype, drug-susceptibility in a sputum sample.

In this pilot study, we evaluated the performance of this portable sequence system for Mtb species identification and detection of genes related to drug resistance, as a means of MDR-TB testing in a diverse set of samples, including DNA isolated from sputum samples and from clinical microbiological isolates.

## 2. Results

### 2.1. Identification of Mtb in Clinical Microbiological Isolates and Sputum Samples

Upon sequencing of a set of DNA obtained from various sources, utilizing the What’s In My Pot (WIMP) app [11] we were able to identify Mtb in all samples evaluated in this study. It was evident that DNA extracted directly from sputa yielded a great number of reads for human DNA (Table 1), with only a few reads for Mtb. In contrast, the presence of reads assigned to *Homo sapiens* in clinical isolates was minimal, and absent in the commercial Mtb genomic DNA.

### 2.2. Identification of Drug Resistance Conferring Mutations in DNA 

We evaluated molecular genome-based drug resistance mutation analysis by sequencing DNA samples using a portable, long-read sequencing platform. Sequenced and analyzed data from Mtb culture isolates and commercially available Mtb genomic DNA showed numerous drug resistance-conferring mutations (Table 2 and Appendix B, Appendix C, Appendix D, Appendix E, Appendix F, Appendix G, Appendix H).

Comparison of the sequencing results of Mtb DNA obtained from direct sputum vs. those of sequencing from Mtb DNA obtained from culture isolates, showed that the amount of reads for Mtb was much higher from the latter samples (Table 2). The higher number of reads translated into a higher number of resistance genes identified.

As pointed out previously, a limitation imposed by sequencing Mtb DNA from sputum samples was the high proportion of human DNA. Despite this relatively low availability of Mtb DNA for sequencing, it proved to be sufficient for obtaining a read coverage that allowed the identification of drug resistance mutations (Table 2).

We then aimed to compare the results obtained for a subset of samples for which whole genome analysis (WGA) data were available. Analyzing the MinION reads in real time with the ARMA pipeline identified a larger number of mutations in genes related to drug resistance, which in some cases included all, or the majority, of those identified by WGS analysis (Table 2). Most of the identified genes had no or poor evidence of their involvement in clinically relevant drug resistance in TB (Appendix B, Appendix C, Appendix D, Appendix E, Appendix F, Appendix G, Appendix H). For those genes with moderate or high-level evidence for drug resistance prediction, some were not supported by the drug susceptibility testing results (Table 2) or were redundant hits. For example, for isolate 6548, isoniazid (INH) resistance was attributed to *katG* (also found in a previous WGA) but also to *inhA* and mutations in the 16S rRNA gene were listed for amikacin, streptomycin and kanamycin resistance independently.

## 3. Discussion

In this study we have evaluated the genomic identification and drug mutation gene profiling of Mtb isolates utilizing the MinION portable sequencer. Our findings endorse the need of further research regarding the practical use of MinION for the detection and characterization of Mtb in clinical isolates and in sputum samples. Our sample set consisted of Mtb genomic DNA obtained by different extraction methods. Recently, low-cost DNA extraction methods for Mtb WGS directly from patient samples have been reported [10], allowing the bypass of laboratory equipment requirements for genomic DNA obtainment.

The portable WGS-based detection system utilized here proved to be fast, relatively inexpensive, with rapid and simple library preparation, and automated real-time analysis tools [10]. The most innovative aspect of this sequencing system is its portability. Its small size and use of a USB port are ideal as they reduce the infrastructure required for WGS sequencing, such as a climate-controlled building, instead requiring only a laptop computer for the system to be operational [9,12]. 

The MinION has several advantages that make it uniquely suited for TB surveillance (Appendix A). Amongst its features, the MinION provides long-read sequencing data, which are ideal for the detection of antimicrobial resistance genes [13], and some authors suggest that this can be achieved even without the need of a high amount of reads [14]. The real time monitoring allows the analysis of metagenomes from complex samples, which could save the 14 days of culture required for drug susceptibility testing in TB. In our set of DNA samples obtained directly from sputum, the presence of host DNA was far more abundant than Mtb DNA, but bacterial DNA could be discriminated and drug resistance related genes were detected, albeit at low sequencing depth. Although identification at the MTBC level provided by Xpert and other fast methods is usually enough for the diagnosis of TB, direct species assignment from sputum samples is an advantage to highlight. Another big challenge in the clinical setting is the bioinformatics analysis, as most clinical labs do not have trained personnel. Real time antimicrobial resistance profiling is indeed, a crucial advantage to highlight. The steps from raw data acquisition to analysis completion are fairly simple and easy to follow in their user-friendly EPI2ME platform [15]. Furthermore, the analysis can be performed in real time even from the moment data acquisition begins, potentially minimizing the results waiting time even more. 

The number of mutations in drug resistance related genes overly surpassed those detected in previous WGA. This may have several explanations. First, a high error rate has been acknowledged as a limitation of Nanopore technology [16], thus, some of these could correspond to sequencing errors, in spite of the overall accuracy of around 90%, according to the automated results. The initial high error rates reported for the MinION [17], have improved over the past few years [18], currently over 95% raw read accuracy and 99.9% consensus read accuracy is achievable. Incorporation of complementary short read sequences [18], and the use of short DNA target sequences, circularized and then amplified via rolling-circle amplification to produce high fidelity accurate repeats [19], are new proposed ways to reduce the error rate. Additionally, recent statistical methods have been reported to aid in the accurate detection of true mutations [20] Long read sequencing has a superior advantage over short read WGS approach, especially in homopolymeric regions where indel is commonly used by bacteria as a drug resistance strategy [21]. Therefore, although the higher number of drug resistance (DR) related genes found in this study using MinION may be due in part a high error rate, it is also reasonable to think that more genes were detected by the long read sequencing compared to the traditional short read WGS. Further investigation is needed to clarify this. Additionally, it would be interesting to follow up on a newer version (R10) of Nanopore’s Flowcell compared to the version used in this study (R9), as improved accuracy with longer barrel and dual reader head in the sequencing pores shall provide better accuracy especially in homopolymer regions. Alternatively, the higher number of detected DR genes by the MinION could correspond to false positive hits detected by the automated ARMA pipeline. The WHO endorses the use of next-generation sequencing analysis for drug-resistance profiling, only for a limited number of genes (*rpoB*, *katG* and *inhA* for first line drugs and *gyrA*, *gyrB*, *rss* and *eis* promoters for second line drugs) and for specific point mutations within them [16]. However, the reference database used by the ARMA pipeline includes genes that lack empirical support for their clinical relevance in TB [16,22]. Almost half of the “hits” corresponded to this category (see Appendix B, Appendix C, Appendix D, Appendix E, Appendix F, Appendix G, Appendix H), indicating that the reference database needs further curation. In addition, some mutations in known resistance conferring genes could correspond to polymorphisms with no functional impact depending on the mutated codon (this is not disclosed in the automated analysis) [16], which could explain the detection of resistance related genes in susceptible isolates. The same could be said for genes like *mtrA* or AAC(2’)-IC, which were detected in 5 out of 7 isolates irrespective of their resistance profile and could correspond to polymorphisms.

Nevertheless, the sensitivity of the MinION sequencing for the detection of drug resistance mutations was good. Isolates 410 and 6548 belong to the extensively studied MDR M strain [23,24] which accumulated resistance to several drugs. The ARMA pipeline detected three of the four drug resistance conferring mutations and an additional mutation in isolate 410, and all six resistance mutations of isolate 6548. Interestingly, the *gidB* mutation, which confers resistance to streptomycin, is not the most frequent among clinical isolates but is characteristic of this cluster and was acquired four decades ago when the expansion of this cluster began [24]. In addition, a rifampicin resistance conferring mutation was found in the metagenome of the sputum sample 1766, which belongs to the Ra cluster, another conspicuous MDR strain of Argentina [25]. These phenotypically confirmed drug-resistance conferring mutations were identified with two to 17 reads depending on the gene, with similar accuracy values. This indicates that although it is usually regarded a critical variable in the analysis of next-generation sequencing data, sequencing depth was not the main constraint in our work. Prompt and accurate information on Mtb strains would have implications for management to minimize transmission of drug-resistant TB and start the most appropriate TB control and anti-TB therapy. Various phylogenetic lineages of the Mtb complex are distributed differently around the world [2]. In Latin America, both drug susceptible and drug resistant TB are mainly related to the Euro–American Lineage [26,27,28,29] and the Beijing strain has a minor impact, in contrast to what is reported in other regions. Drug resistance databases mostly rely on the genome H37Rv strain. It is interesting to challenge this sequencing system with samples sets with diverse genetic backgrounds like ours to assess its impact in the performance.

Overall, our findings indicate that the improvements in the future should focus on: (1) recovering higher number of reads corresponding to Mtb from sputa; (2) lowering MinION sequencing error rates; (3) improving the drug-resistance conferring mutation detection algorithms for automated analysis and (4) curating the reference database to include only those hits that have a strong correlation with Mtb drug resistance phenotype.

Although our data relies on a short number of DNA samples, our findings suggest that this portable DNA sequencing system could be effective in reducing time and providing information on Mtb genotype and drug-susceptibility from direct sputum samples. As larger studies—evaluating parameters such as the minimal number of reads for a complete reliable drug susceptibility profiling, optimization in software and database accuracy for the prediction of new drug resistance genes, and reduction in false positive drug detection—are conducted, this system could potentially revolutionize current TB testing procedures, especially in genomic surveillance for MDR-TB in the clinical setting.

## 4. Materials and Methods

### 4.1. Mtb Genomic DNA 

Mtb genomic DNA, strain HN878 was acquired through bei resources (NR-14867). Genomic bacterial DNA extracted from four laboratory cultured Mtb isolates from sputum samples was also utilized (Table 1). These correspond to a strain belonging to the Beijing lineage, a strain belonging to the Latin American and Mediterranean (LAM) lineage [27], and two closely related strains belonging to the Haarlem lineage, the so called M strain (isolate 6548), and an M strain variant (isolate 410). Genomic DNA was also extracted directly from 2 sputum samples from pulmonary TB patients with positive bacilloscopy scored through correspondent acid-fast bacteria (AFB) smears. These latter samples included a strain susceptible to the first line drugs INH, RIF, STR, EMB (sample 2836) and an Ra strain (sample 1766) which along with the M strain constitute the most prevalent MDR clusters in Argentina [30]. MPure^TM^ DNA Extraction Kit (MP Biomedical), as well as inactivation and lysis by sonication protocol [27] were used for bacterial DNA extraction, except for DNA from strain HN878, which used a delipidation method, followed by lysozyme, RNase, SDS and proteinase digestion [31]. 

### 4.2. Whole Genome Sequencing (WGS) Data

Whole genome sequencing (WGS) with Illumina was available for isolates 6548 and 410 [23] and for representative isolates of the Ra cluster for comparison with sputum sample 1766 [22]. WGS data from Mtb was obtained by eliminating human DNA sequences utilizing “Read Until” approach (OMICtools) for target sequencing [32]. Mtb identification was performed once the metagenome was obtained.

### 4.3. MinION DNA Sequencing and Resistance Gene Identification

DNA sample libraries were constructed using Rapid Sequencing Kit (ONT, Littlemore, UK), and sequencing was conducted on MinION-compatible R9.4 flow cells (ONT, UK). Primary data acquisition was done using MinKNOW, the operating software that drives nanopore sequencing devices. Raw data were processed for basecalling via Albacore. Data were then further processed using the cloud-based data analysis platform EPI2ME [15]. Microbial species identification was done using the What’s In My Pot (WIMP) analysis workflow [11], and detection of mutations conferring antibiotic drug resistance was done through the real time antimicrobial resistance mapping application (ARMA) [33]. 

## 5. Conclusions

In this study we have evaluated the genomic identification and drug mutation gene profiling of Mtb isolates utilizing the MinION portable sequencer The approach was highly sensitive, being able to detect almost all described drug resistance conferring mutations based on previous whole genome sequencing analysis. Our findings are supportive of the practical use of this system as a suitable method for the detection of antimicrobial resistance genes, and effective in providing Mtb genomic information. Future improvements in the error rate through statistical analysis, drug resistance prediction algorithms and reference databases would make this a platform suited for the clinical setting. The small size, relatively inexpensive cost of the device, as well as its rapid and simple library preparation protocol and analysis, make it an attractive option for settings with limited laboratory infrastructure. 

## Figures and Tables

**Table 1 antibiotics-09-00548-t001:** Microbial identification through sequencing DNA analysis. LAM: Latin American and Mediterranean, WIMP: What’s In My Pot.

Samples	Source	WIMP Species Identification	
		Bacteria	Eukaryota
		*Mycobacterium tuberculosis*	*Homo sapiens*
HN878	Genomic DNA (bei-resources)	28,090	0
LAM	Clinical isolate	8066	8
Beijing	Clinical isolate	1772	2
410	Clinical isolate	6736	376
6548	Clinical isolate	9664	39
1766	Sputum	53	420,062
2836	Sputum	16	56,450

Data shown as cumulative reads.

**Table 2 antibiotics-09-00548-t002:** Mutations observed in drug resistance (DR) related genes through MinION and previous whole genome analysis (WGA) sequencing analysis.

Samples	Source	DR Related Genes			Phenotypic Resistance Validation
		Minion	WGA ^a^	Detected by both Systems	
HN878	Genomic DNA (bei-resources)	33	0	-	-
LAM	Clinical isolate	20	N/A	N/A	N/A
Beijing	Clinical isolate	32	N/A	N/A	N/A
410	Clinical isolate	34	5	4	3/4 ^b^
6548	Clinical isolate	29	6	6	6/6
1766	Sputum	5	3	1	1/3
2836	Sputum	3	N/A	N/A	N/A

N/A: not available. ^a^ Ref 24 ^a^ One of the concordant hits among the sequencing experiments corresponded to the *embB* gene, which has poor evidence of a correlation with phenotypic drug resistance; in accordance, the strain was susceptible to ethambutol in vitro. Full description of the results is available in Appendix B, Appendix C, Appendix D, Appendix E, Appendix F, Appendix G, Appendix H.

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
