# Peer review of "Genetic Identification and Drug-Resistance Characterization of Mycobacterium tuberculosis Using a Portable Sequencing Device. A Pilot Study"

_antibiotics, 2020, doi:10.3390/antibiotics9090548_

Round 1

Reviewer 1 Report

In clinical examinations for tuberculosis, traditional phenotypic drug susceptibility testing for Mycobacterium tuberculosis (Mtb) is costly and time consuming, whereas PCR-based methods are limited to selected target loci. The authors utilized MinION to analyze drug-resistance conferring mutations in Mtb strains and reported that the method is sensitive and lead to detection of almost all described drug resistant conferring mutations. It is concluded that MinION is an attractive option for this purpose in the manuscript.

Although the results are considered positive by the authors, their elucidation is doubtful. Much more mutations in drug resistant-related genes were observed by MinION than Illumina (Table 2), which must be caused by a high error rate in sequencing by MinION. The manuscript might provide readers with useful information that publish negative aspects of MinION in genome analysis of Mtb strains. But, since the high error rate by MinION is generally known, results in this manuscript are so predictable that it does not provide any new findings although it is a pilot study. Low accuracies shown in Annexes are problematic in the purpose.

Author Response

Reviewer 1

In clinical examinations for tuberculosis, traditional phenotypic drug susceptibility testing for Mycobacterium tuberculosis (Mtb) is costly and time consuming, whereas PCR-based methods are limited to selected target loci. The authors utilized MinION to analyze drug-resistance conferring mutations in Mtb strains and reported that the method is sensitive and lead to detection of almost all described drug resistant conferring mutations. It is concluded that MinION is an attractive option for this purpose in the manuscript.

Although the results are considered positive by the authors, their elucidation is doubtful. Much more mutations in drug resistant-related genes were observed by MinION than Illumina (Table 2), which must be caused by a high error rate in sequencing by MinION. The manuscript might provide readers with useful information that publish negative aspects of MinION in genome analysis of Mtb strains. But, since the high error rate by MinION is generally known, results in this manuscript are so predictable that it does not provide any new findings although it is a pilot study. Low accuracies shown in Annexes are problematic in the purpose.

-Despite MinION’s accuracy having improved over the past few years (Rang et al 2018), it is still lower than short reads systems like Illumina. The average accuracy for each of the strains we sequenced ranged between 85 to 91%.

We have included a series of statements in the Discussion to address these issues, as follows:

[Lines 150 -157]: “…a high error rate has been acknowledged as a limitation of Nanopore technology (16), thus some of these could correspond to sequencing errors, in spite of the overall accuracy of around 90% according to the automated results. The initial high error rates reported for the MinION (17), have improved over the past few years (18), currently over 95% raw read accuracy and 99.9% consensus read accuracy achievable. Incorporation of complementary short read sequences (18), and the use of short DNA target sequences, circularized and then amplified via rolling-circle amplification to produce high fidelity accurate repeats (19), are new proposed ways to reduce the error rate. Also, recent statistical methods have been reported to aid in the accurate detection of true mutations (20).”

Reviewer 2 Report

Manuscript ID: antibiotics-867035  REVIEW REPORT

This is a report for the manuscript entitled “Genetic Identification and Drug-Resistance Characterization of Mycobacterium tuberculosis Using a Portable Sequencing Device. A Pilot Study”.

The manuscript presents and discusses the usability of the MinION sequencing instrument for a rapid and accurate identification and characterization of drug-resistant Mycobacterium tuberculosis (Mtb). This was a pilot study, as recognized by the authors, considering the few DNA samples (seven) under analysis: one DNA acquired commercially from a Mtb strain (HN878), and six DNAs extracted by the authors (four from clinical microbiological isolates, and two from sputum samples).

The results are very impressive and allowed to conclude that this is a promising approach, although still needing improvements suggested by the results themselves.

Sputum samples shown to be the more problematic, mainly due to an excessive amount of human DNA (expressed in 420,062 and 56,450 reads); however, albeit the few number of Mtb reads (16 and 53) in these samples, it was possible, in one of the two samples, to validate one read (in three expected and in five detected) as a drug related (DR) gene. These were the worse results, as the sensitivity was much better for the DNA extracted from the two clinical isolates with known DR genes (3/4 and 6/6), based on data from previous whole genome sequencing analysis (WGA).

The authors also evidenced the consistent obtention of more reads related to DR genes, than those expected by WGA. These results were related to the accuracy of the whole approach, which was addressed and discussed at the different levels (e.g. error rate of the nanopore technology, data not disclosed in the automated analysis of the reads).

Globally, the experimental approaches were adequate, the results are clear and impressive (well-documented and supported by supplementary material), and the discussion is comprehensive and concise.

More specific comments:
Considering that this manuscript could be improved, more detailed considerations are presented below:

Abstract: Should also provide information about the limitations to overcome before assuming MinION as a highly reliable solution.

Introduction: The scope of the study and its originality/relevance are well-addressed: to challenge MinION performance in providing information on Mtb genotype and drug-susceptibility, when DNA is extracted from patient sputum samples.

Results and Discussion: As expected from a pilot study, there are few results and even very few (only a couple, and just one with WGS data available) concerning the main point of this study: use of patient sputum samples. This is its main weakness, that was evidenced by main differences in response between the two samples. These results would conduct to the impossibility to conclude that MinION performed well with these samples. The authors recognized this endorsing for “the need of further research…” (lines 126 – 128).

The clinical samples (four but only two of them had available WGS data) did not provide enough results to give good reasons for the choice of MinION, as also recognized in the same sentence in lines 126 – 128.

As referred in my general comment another relevant result that evidences the limitations of the data provided by MinION is “the number of mutations in drug resistance related genes overly surpassed those detected in previous WGA” (lines 151 – 152). The manuscript is very well documented about this subject, including informative supplementary tables. And the authors address, not so extensively as would be desired, possible origin/s and eventual solutions for these results (lines 153 – 161; 176 - 178).

On the other end, this study allowed to state that the sensitivity of the MinION was good in the detection of drug resistance mutations (lines 162 – 170). This is the part of the Discussion that could be further developed to validate the results in terms of field application.

Maybe good is not enough but it is OK as a starting point. And when planned for situations where the alternative is “nothing” or “too late”, the strategy of exploring all the possibilities of this device is perfectly acceptable.
The above explanations are, in my opinion, the reason for the acceptance of this manuscript. Not for relevant and decisive acquaintances, but much more because there are results to share and may be considered as starting points for others. And, why not, for these same authors?

Author Response

Reviewer 2

This is a report for the manuscript entitled “Genetic Identification and Drug-Resistance Characterization of Mycobacterium tuberculosis Using a Portable Sequencing Device. A Pilot Study”.

The manuscript presents and discusses the usability of the MinION sequencing instrument for a rapid and accurate identification and characterization of drug-resistant Mycobacterium tuberculosis (Mtb). This was a pilot study, as recognized by the authors, considering the few DNA samples (seven) under analysis: one DNA acquired commercially from a Mtb strain (HN878), and six DNAs extracted by the authors (four from clinical microbiological isolates, and two from sputum samples).

The results are very impressive and allowed to conclude that this is a promising approach, although still needing improvements suggested by the results themselves.

Sputum samples shown to be the more problematic, mainly due to an excessive amount of human DNA (expressed in 420,062 and 56,450 reads); however, albeit the few number of Mtb reads (16 and 53) in these samples, it was possible, in one of the two samples, to validate one read (in three expected and in five detected) as a drug related (DR) gene. These were the worse results, as the sensitivity was much better for the DNA extracted from the two clinical isolates with known DR genes (3/4 and 6/6), based on data from previous whole genome sequencing analysis (WGA).

The authors also evidenced the consistent obtention of more reads related to DR genes, than those expected by WGA. These results were related to the accuracy of the whole approach, which was addressed and discussed at the different levels (e.g. error rate of the nanopore technology, data not disclosed in the automated analysis of the reads).

Globally, the experimental approaches were adequate, the results are clear and impressive (well-documented and supported by supplementary material), and the discussion is comprehensive and concise.

-We thank the reviewer for his praising and clear understanding of our findings.

More specific comments:
Considering that this manuscript could be improved, more detailed considerations are presented below:

Abstract: Should also provide information about the limitations to overcome before assuming MinION as a highly reliable solution.

-The following has been added at the end of the Abstract [Lines 24-26]: “Future improvements in the error rate through statistical  analysis, drug resistance prediction algorithms and reference databases would make this a platform suited for the clinical setting.

Introduction: The scope of the study and its originality/relevance are well-addressed: to challenge MinION performance in providing information on Mtb genotype and drug-susceptibility, when DNA is extracted from patient sputum samples.

-We appreciate Reviewer’s comment.

Results and Discussion: As expected from a pilot study, there are few results and even very few (only a couple, and just one with WGS data available) concerning the main point of this study: use of patient sputum samples. This is its main weakness,that was evidenced by main differences in response between the two samples. These results would conduct to the impossibility to conclude that MinION performed well with these samples. The authors recognized this endorsing for “the need of further research…” (lines 126 – 128).

-Indeed, the limited amount of samples and WGA data for comparison, are the main limitations of our pilot study. Nevertheless, as the Reviewer points out later on, this is a good starting point for us and other researchers in the field.

The clinical samples (four but only two of them had available WGS data) did not provide enough results to give good reasons for the choice of MinION, as also recognized in the same sentence in lines 126 – 128.

As referred in my general comment another relevant result that evidences the limitations of the data provided by MinION is “the number of mutations in drug resistance related genes overly surpassed those detected in previous WGA” (lines 151 – 152). The manuscript is very well documented about this subject, including informative supplementary tables. And the authors address, not so extensively as would be desired, possible origin/s and eventual solutions for these results (lines 153 – 161; 176 - 178).

- We have included a series of statements in the Discussion to address these issues, as follows:

[Lines 150 -157]: “…a high error rate has been acknowledged as a limitation of Nanopore technology (16), thus some of these could correspond to sequencing errors, in spite of the overall accuracy of around 90% according to the automated results. The initial high error rates reported for the MinION (17), have improved over the past few years (18), currently over 95% raw read accuracy and 99.9% consensus read accuracy achievable. Incorporation of complementary short read sequences (18), and the use of short DNA target sequences, circularized and then amplified via rolling-circle amplification to produce high fidelity accurate repeats (19), are new proposed ways to reduce the error rate. Also, recent statistical methods have been reported to aid in the accurate detection of true mutations (20).”

We did not find a correlation between the number of reads and accuracy. Hits that were concordant with the mutations detected by Illumina (blue dots) had variable levels of accuracy (average = 89.1% [75.3% - 92.8%], global average = 87.9%), and many false-positive hits had >50 reads, indicating that this variable was not the main limitation of our results. Accordingly, we added the following:  

[Lines 182-185]:“These phenotypically confirmed drug-resistance conferring mutations were identified with 2 to 17 reads depending on the gene, and accuracy values were comparable to the global average. This indicates that although it is usually regarded a critical variable in the analysis of next-generation sequencing data, sequencing depth was not the main constraint in our work.”

On the other end, this study allowed to state that the sensitivity of the MinION was good in the detection of drug resistance mutations (lines 162 – 170). This is the part of the Discussion that could be further developed to validate the results in terms of field application.

We agree with the reviewer and the following paragraph has been added to deepen the discussion:

[Lines 157-174 ]:” Long read sequencing has a superior advantage over short read WGS approach especially in homopolymeric regions where indel is commonly used by bacteria as a drug resistance strategy (21). Therefore, although the higher number of DR related genes found in this study using MinION may be due in part a high error rate, it is also reasonable to think that more genes were detected by the long read sequencing compared to the traditional short read WGS. Further investigation is needed to clarify this. Also, it would be interesting to follow up on a newer version (R10) of Nanopore’s flowcell compared to the version used in this study (R9), as improved accuracy with longer barrel and dual reader head in the sequencing pores shall provide better accuracy especially in homopolymer regions. Alternatively, the higher number of detected DR genes by the MinION could correspond to false positive hits detected by the automated ARMA pipeline. The WHO endorses the use of next-generation sequencing analysis for drug-resistance profiling only for a limited number of genes (rpoB, katG and inhA for first line drugs and gyrA, gyrB, rss and eis promoter for second line drugs) and  for specific point mutations within them (22).  However, the reference database used by the ARMA pipeline includes genes that lack empirical support for their clinical relevance in TB (16, 23). Almost half of the “hits” corresponded to this category (see Annex I to VII), indicating that the reference database needs further curation. In addition, some mutations in known resistance conferring genes could correspond to polymorphisms with no functional impact depending on the mutated codon (this is not disclosed in the automated analysis) (22), which could explain the detection of resistance related genes in susceptible isolates. The same could be said for genes like mtrA or AAC(2’)-IC which were detected in  5/7 isolates irrespective of their resistance profile and could correspond to polymorphisms. “

In addition, the possible future improvements are pointed as follows in the current version:

[Lines 193-196]:“Overall, our findings indicate that the improvements in the future should focus on: 1) recovering higher number of reads corresponding to Mtb from sputa, 2) lowering MinION sequencing error rates, 3) improving the drug-resistance conferring mutation detection algorithms for automated analysis and 4) curating the reference database to include only those hits that have a strong correlation with Mtb drug resistance phenotype.”

Both low sensitivity and specificity of molecular tests have negative impact on the patient treatment according to the WHO (https://www.who.int/tb/WHOPolicyStatementSLLPA.pdf), thus it is important to improve both. Particularly, false positive results can exclude a drug that is actually working from the treatment regimen. Although points 3 and 4 can be addressed with manual analysis of the obtained sequences, it is beyond our possibilities to improve these in the automated workflow.

Maybe good is not enough but it is OK as a starting point. And when planned for situations where the alternative is “nothing” or “too late”, the strategy of exploring all the possibilities of this device is perfectly acceptable.
The above explanations are, in my opinion, the reason for the acceptance of this manuscript. Not for relevant and decisive acquaintances, but much more because there are results to share and may be considered as starting points for others. And, why not, for these same authors?

Reviewer 3 Report

While authors correctly note that "Comparison with phenotypic resistance further confirmed that most of the findings of the MinION analysis were not supported by the drug susceptibility testing results" (lines 117-119) they do not discuss this very important limitation to the practical usability of MinIon system. The notion that "In any case, a detailed manual analysis is needed to address these questions" (line 161) further underlines the need to better research actual field applicability.
It is not clear why two different DNA extraction methods were used (MPureTM DNA Extraction Kit for most of the strains and delipidation method, followed by lysozyme, RNase, SDS and proteinase digestion for the strain HN878). As the extraction method may significantly affect both quantity and quality of the DNA extracted such difference should either be better explained or comparisons to the strain HN878 made with great caution.

Minor language and spelling errors in the manuscript should be corrected, i.e. on line 179 the beginning of the sentence "Although our data relies oa short number of DNA samples ..." should read "Although our data relies on a small number of DNA samples..."

Author Response

While authors correctly note that "Comparison with phenotypic resistance further confirmed that most of the findings of the MinION analysis were not supported by the drug susceptibility testing results" (lines 117-119) they do not discuss this very important limitation to the practical usability of MinIon system. The notion that "In any case, a detailed manual analysis is needed to address these questions" (line 161) further underlines the need to better research actual field applicability.

We modified the cited passages for more clarity:

In Results, subheading “Identification of drug resistance conferring mutations in DNA”, [Lines 107-112]: “Most of the identified genes had no or poor evidence of their involvement in clinically relevant drug resistance in TB (Annex I-VII). For those genes with moderate or high level evidence for drug resistance prediction, some were not supported by the drug susceptibility testing results (Table 2) or were redundant hits. For example, for isolate 6548, INH resistance was attributed to katG (also found in a previous WGA) but also to inhA and mutation in 16S rRNA gene were listed for amikacin, streptomycin and kanamycin resistance independently.”

We have also replaced the statement “In any case, a detailed manual analysis is needed to address these questions. “, with the following: “[Lines 154 -157]: “Incorporation of complementary short read sequences (18), and the use of short DNA target sequences, circularized and then amplified via rolling-circle amplification to produce high fidelity accurate repeats (19), are new proposed ways to reduce the error rate. Also, recent statistical methods have been reported to aid in the accurate detection of true mutations (20).”This paragraph was re-written to deepen the discussion. A summary of the possible improvements to address in the future was also included.

It is not clear why two different DNA extraction methods were used (MPureTM DNA Extraction Kit for most of the strains and delipidation method, followed by lysozyme, RNase, SDS and proteinase digestion for the strain HN878). As the extraction method may significantly affect both quantity and quality of the DNA extracted such difference should either be better explained or comparisons to the strain HN878 made with great caution.

-Genomic DNA from strain HN878 was acquired through bei resources. Thus, the DNA extraction method they used is completely out of our control. We just provided the reference mentioned in the product insert.

Minor language and spelling errors in the manuscript should be corrected, i.e. on line 179 the beginning of the sentence "Although our data relies oa short number of DNA samples ..." should read "Although our data relies on a small number of DNA samples..."

-The mistake has been amended.

Reviewer 4 Report

The goal of the manuscript by Cervantes et al. “Genetic identification and drug resistance characterization of Mycobacterium tuberculosis using a portable sequencing device” is to demonstrate usefulness of MinION-operated long-read DNA sequencing in clinical management of tuberculosis. To do so, MinION technology should enable facile, on-site determination of M. tuberculosis (Mtb) genomic DNA presence, as well as the presence of anti-TB drug resistance conferring mutations. To showcase MinION capability in this matter, the authors sequenced genomic DNA extracted from (i) HN878 strain; (ii) DNA obtained from M. tuberculosis isolates cultured from sputum samples, one of which belonged to the Beijing lineage, one to LAM and two to the Haarlem lineage; (iii) DNA extracted directly from the sputum of two pulmonary TB patients. Comparison between the samples revealed that in the samples taken directly from patients’ sputum human DNA greatly outnumbers Mtb DNA resulting with low number of reads of Mtb DNA, and consequently, lower number of mutations identified in drug-resistance related genes. Comparison between sequences obtained through MinION vs those obtained through Illumina sequencing technology reveals a much larger number of mutations identified by MinION, especially in case of clinical isolates.

While this work is based on sound experiments, I find that the major criticism would be the following: knowing that nanopore sequencing still suffers from high error rates (which renders the detection of true variants difficult) what other analyses would the authors undertake to remove false positives/negatives. Statistical methods to aid this have been reported (such as AssociVar (Harel et al., 2019)).

Minor remarks:

line 22: perhaps this sentence could be written in a clearer manner; like this, it reads that the “genes…identified Mtb species”.

lines 98 and 121: Table titles should be completed.

line179: on, not “oa”.

Author Response

Reviewer 4

The goal of the manuscript by Cervantes et al. “Genetic identification and drug resistance characterization of Mycobacterium tuberculosis using a portable sequencing device” is to demonstrate usefulness of MinION-operated long-read DNA sequencing in clinical management of tuberculosis. To do so, MinION technology should enable facile, on-site determination of M. tuberculosis (Mtb) genomic DNA presence, as well as the presence of anti-TB drug resistance conferring mutations. To showcase MinION capability in this matter, the authors sequenced genomic DNA extracted from (i) HN878 strain; (ii) DNA obtained from M. tuberculosis isolates cultured from sputum samples, one of which belonged to the Beijing lineage, one to LAM and two to the Haarlem lineage; (iii) DNA extracted directly from the sputum of two pulmonary TB patients. Comparison between the samples revealed that in the samples taken directly from patients’ sputum human DNA greatly outnumbers Mtb DNA resulting with low number of reads of Mtb DNA, and consequently, lower number of mutations identified in drug-resistance related genes. Comparison between sequences obtained through MinION vs those obtained through Illumina sequencing technology reveals a much larger number of mutations identified by MinION, especially in case of clinical isolates.

While this work is based on sound experiments, I find that the major criticism would be the following: knowing that nanopore sequencing still suffers from high error rates (which renders the detection of true variants difficult) what other analyses would the authors undertake to remove false positives/negatives. Statistical methods to aid this have been reported (such as AssociVar (Harel et al., 2019)).

-The Reviewer is correct and we have addressed this limitation in the Discussion section stating that:

[Lines 150 -157]: “…a high error rate has been acknowledged as a limitation of Nanopore technology (16), thus some of these could correspond to sequencing errors, in spite of the overall accuracy of around 90% according to the automated results. The initial high error rates reported for the MinION (17), have improved over the past few years (18), currently over 95% raw read accuracy and 99.9% consensus read accuracy achievable. Incorporation of complementary short read sequences (18), and the use of short DNA target sequences, circularized and then amplified via rolling-circle amplification to produce high fidelity accurate repeats (19), are new proposed ways to reduce the error rate. Also, recent statistical methods have been reported to aid in the accurate detection of true mutations (20).”

 [Lines 193-196]: “Overall, our findings indicate that the improvements in the future should focus on: 1) recovering higher number of reads corresponding to Mtb from sputa, 2) lowering MinION sequencing error rates, 3) improving the drug-resistance conferring mutation detection algorithms for automated analysis and 4) curating the reference database to include only those hits that have a strong correlation with Mtb drug resistance phenotype.”

As explained to reviewer 2, we consider that possible improvements in points 2 to 4 are beyond our possibilities as end users of this technology.

Minor remarks:

line 22: perhaps this sentence could be written in a clearer manner; like this, it reads that the “genes…identified Mtb species”.

-The sentence has been modified to: “Data analysis platform EPI2ME and antibiotic-resistance analysis using ARMA workflow, identified Mtb species as well as extensive resistance gene profiles.

lines 98 and 121: Table titles should be completed.

-Table titles now read:

Table 1. Microbial identification through sequencing DNA analysis

Table 2. Mutations observed in drug resistance related genes through MinION and WGA sequencing analysis

line179: on, not “oa”.

-The mistake has been corrected

Round 2

Reviewer 1 Report

The revised manuscript should be also rejected for publication. I think that the authors are not able to consider the results correctly. The conclusion is that MinION is not useful for the aim because of frequent sequencing errors. Without this study, readers have known it. To determine the presence of drug resistance-conferring mutations, accurate sequencing methods should be employed.

L26-27. The sentence is inappropriate. Their findings never support the practical use. MinION is not a suitable method for the detection because of its high error rate.

L28. The error rate is not through statistical analysis but derived from low sequencing accuracy of MinION.

Table 1. Why did reads of Homo sapience appear from samples of clinical isolates? ‘Isolate’ means a purified strain. As reads of Homo sapience are observed from isolated strains, the experimental skill of the authors is not enough, and the results are doubtful.

Table 2. More mutations were detected in the results by MinION. Most of them must be derived from sequencing errors by MinION.

L151. The 99.9% accuracy is not enough for the aim. Although 99.9% consensus read accuracy is available, accuracies in this study are about 90%.

L158-160. The elucidation is inappropriate. If more genes were really detected, please show the list of detected genes (not only mutated genes but also all) by MinION and Illumina and then compare them. I guess that the traditional short read WGS can also detect almost all genes. If sequences are different between by MinION and Illumina, those by Illumina are more reliable.

L160. Further investigation is not needed. Without it, MinION is concluded to be not useful for the purpose such as detection of mutation because of low sequencing accuracy. Without the present work by the authors, such the results are easily predictable, and this work lacks any significance in this journal.

L160-163. If the results by R10 are comparable to those by Illumina, the paper may be accepted for peer-reviewing.

L163-165. If so, do not use the pipeline.

L169-175. What they mean are unclear. The descriptions are not logical at all and hardly acceptable for readers of this journal.

L176-178. The sensitivity is not good because most of the mutations must be derived from sequencing errors.

Annex. The accuracies are too low. Without accuracy of 100%, the results must not be published in the journal.

Author Response

The manuscript "Genetic Identification and Drug-Resistance Characterization of Mycobacterium tuberculosis Using a Portable Sequencing Device. A Pilot Study" is on an important public health topic. Nevertheless, the following issues should be addressed:

- Line 33: Consider drug resistance (DR). The abbreviation is used in the texto and tables but was not defined.

We think keeping “drug resistance” in the narrative makes the article easier to read.

- Table 1. Please address the issue raised by reviewer 1 "Table 1. Why did reads of Homo sapience appear from samples of clinical isolates? ‘Isolate’ means a purified strain. As reads of Homo sapience are observed from isolated strains, the experimental skill of the authors is not enough, and the results are doubtful." If isolates (meaning strains) were used the presence of host DNA is not expected. Nevertheless, if DNA was extracted directly from clinical samples (sputum or others) the presence of host DNA is expected. Please explain.

-There are two possible explanations for observing hits for Homo sapiens. The first one is contamination, as it has been observed in other reports using MinION (Fauver et al. Scientific Reports 2019 9:19521 De novo assembly of the Brugia malayi genome using long reads from a single MinION flowcell). In fact, contamination with foreign DNA of “pure” cultures is frequently observed, even leading to annotation errors (Breitwieser, Genome research, 2019, 10.1101/gr.245373.118) and their impact in the final analysis has been studied recently (Goig, BMC biology, 2020, 10.1186/s12915-020-0748-z).

The other one is the fact that these are blasts hits that can appear in shared genomic sequences and are misclassified. Although table 1 shows hits for Homo sapiens, there were several other organisms that also appeared as well (with a lower number of hits). If an increase in the number of shared sequencing between organisms is greater for sequences that are mutated sequences is unknown, but possible.

In either case, although wet lab work can certainly be improved, computational approaches can overcome the negative impact that these spurious sequences might have, as described by Goig et al. in their recent paper.

- Lines 137-139: "In our set of DNA samples obtained directly from sputum, the presence of human DNA was much greater than that one of Mtb, but bacterial DNA could be discriminated and drug resistance related genes were detected albeit at low sequencing depth." Please consider revising the part "DNA was mucher greater than". I assume the authors meant that host DNA was more abundant than Mtb DNA.

- We have changed the statement, and it now reads: the presence of host DNA was far more abundant than Mtb DNA”.